# Impact of postoperative macular comorbidity on visual outcomes after Descemet's membrane endothelial keratoplasty: A multicenter analysis

**Naoya Nakagawa**[1☉], **Ami Igarashi**[1☉], **Hideaki Yokogawa**[2], **Akira Kobayashi**[2],
**Tomomi Higashide**[2], **Satoru Yamagami**[1], **Takahiko Hayashi**[1*]

1 Department of Ophthalmology, Department of Visual Sciences, Nihon University School of Medicine, Tokyo, Japan, 2 Department of Ophthalmology, Kanazawa University Hospital, Kanazawa University Graduate School of Medical Science, Kanazawa, Ishikawa, Japan

☉ These authors contributed equally to this work.
* takamed@gmail.com

## Abstract

The aim of this study was to identify the clinical factors associated with postoperative visual acuity following Descemet's membrane endothelial keratoplasty (DMEK), with emphasis on the impact of macular diseases. This retrospective multicenter study included consecutive eyes that underwent DMEK between March 1, 2011, and June 30, 2022, and had available optical coherence tomography findings. Eyes with other ocular diseases causing visual loss were excluded. Overall, 77 eyes of 66 patients were included in the study. Univariable and multivariable regression analyses were performed to identify the predictors of best-corrected visual acuity (BCVA) at the final follow-up. Worse preoperative visual acuity and macular comorbidity were identified as independent predictors of poorer postoperative BCVA. Among macular patholo-gies, cystoid macular edema was most strongly correlated with reduced vision in the univariable analysis (β = 0.195; p = 0.016). In the subgroup analysis, worse preopera-tive visual acuity and macular comorbidity remained significant predictors (p < 0.001 and p = 0.023, respectively). Other comorbidities such as epiretinal membrane and wet age-related macular degeneration were not significant predictors. Worse pre-operative visual acuity and macular comorbidity are independent risk factors for suboptimal visual recovery after DMEK. These findings highlight the importance of comprehensive preoperative retinal assessment and individualized postoperative management to optimize visual outcomes.

## Introduction

Descemet's membrane endothelial keratoplasty (DMEK) is the preferred technique for endothelial disorders because of its low rejection rate and good visual recovery attributed to low interface haze and higher-order aberrations [1–5]. Therefore, in

**Data availability statement:** All relevant data are within the paper and its Supporting Information file.

**Funding:** The author(s) received no specific funding for this work.

**Competing interests:** The authors have declared that no competing interests exist.

cases of poor vision after DMEK, considerations extend beyond corneal factors, including macular comorbidity.

Cystoid macular edema (CME), including Irvine–Gass syndrome, is a common complication following intraocular surgery, especially after cataract surgery [6–7]. CME is also considered an important factor in triggering visual loss in patients undergoing endothelial corneal transplantation. Previous investigations have indicated a CME incidence rate of 5.6%–15.6% following DMEK [8–10]. Notably, a previous study underscored the association between cataract surgery and an increased risk of developing late-stage age-related macular degeneration (AMD) [11]. Although the incidence of AMD following DMEK remains unverified, because cataract surgery is a prevalent cause of bullous keratopathy (BK) in Japanese eyes [12], AMD should also be considered following DMEK. While there are reports of CME development after DMEK, there are no reports comparing it with other macular comorbidities.

In this study, we investigated the incidence of macular comorbidities, including CME, wet AMD, and epiretinal membrane (ERM), after DMEK surgery. In addition, the association between postoperative corrected visual acuity and macular comorbidity was evaluated.

## Methods

### Study design

The study population comprised patients who underwent DMEK at Yokohama Minami Kyosai Hospital, Kikuna Yuda Eye Clinic, and Kanazawa University Hospital between March 1, 2011, and June 30, 2022. The data were accessed for research purposes between May 1, 2023, and July 31, 2023. Only patients with available optical coherence tomography (OCT) records were included in the analysis. Patients with a history of other diseases that may cause vision loss, such as amblyopia or glaucoma, were excluded from the study. This study was approved by the Ethical Review Board of Yokohama Minami Kyosai Hospital, Kikuna Yuda Eye Clinic, Kanazawa University Hospital, and Nihon University School of Medicine (approval no: RK-210807–1). It adhered to the tenets of the Declaration of Helsinki. All patients provided informed consent in the form of an opt-out on the website. Those who rejected participation were excluded.

### Surgical procedure and postoperative treatment

Standardized DMEK was performed by a single experienced surgeon (TH) as described previously. The pre-stripped donor tissue was carefully cut to an estimated size (approximately 8.0 mm) using a vacuum punch (Moria Japan, Tokyo, Japan) and stained with 0.1% Brilliant Blue G dye. The host's Descemet membrane was removed using a reverse Sinsky hook under air infusion, and a DMEK graft was inserted through the anterior chamber using an intraocular lens (IOL) inserter (WJ-60M; Santen, Osaka, Japan). After the graft was unfolded, 20% SF6 gas was used as a tamponade for graft attachment. Peripheral iridectomy was performed using a 25-G vitreous cutter (Stellaris PC Vitrectomy System; Bausch + Lomb, Laval, Canada). At the end of surgery, 0.4 mg betamethasone (Rinderon; Shionogi, Osaka, Japan) was

administered through subconjunctival injection. All patients were instructed to remain in a supine position for several days. Postoperative treatment included 1.5% levofloxacin (Cravit; Santen), betamethasone (Sanbetason; Santen), and 2% rebamipide ophthalmic solution (Mucosta; Otsuka, Tokyo, Japan) four times daily for three months and was subsequently tapered. Topical tropicamide was not included in the postoperative regimen.

In addition to the standard ophthalmologic examination, anterior segment OCT (SS1000, Tomey, Aichi, Japan) was performed at the follow-up visit.

### Diagnosis and management of macular diseases

Postoperative CME was defined as intraretinal fluid spaces or subretinal fluid in the foveal region, or central macular thickness (central 1000 μm) > 300 μm on Spectralis HRA imaging [13]. Patients with diabetes and diabetic macular edema were not included. Wet (exudative or neovascular) AMD was defined as the presence of a serous or hemorrhagic retinal pigment epithelium detachment, a subretinal neovascularization, subretinal hemorrhage, or peri-retinal fibrous scar [11]. ERM was defined as a fibro-cellular proliferation on the inner retinal surface of the macular area.

When CME was diagnosed, topical bromfenac (Bronuck; Senju, Osaka, Japan) was administered two times per day for two months and topical betamethasone (Sanbetason; Santen, Osaka, Japan) four times for two months. When CME did not resolve with treatment, sub-Tenon injection of triamcinolone acetonide (MaQaid; Wakamoto, Tokyo, Japan) was administered. A fixed treat-and-extend anti-vascular endothelial growth factor (VEGF) regimen was used for wet AMD treatment. Aflibercept 2 mg (Eylea, 40 mg/ml, Bayer Pharma AG, Leverkusen, Germany) was used as anti-VEGF. The final visual acuity was measured after the resolution of CME following treatment.

### Statistical methods

Patient demographics and baseline clinical characteristics were summarized using descriptive statistics. Continuous variables are expressed as the mean ± standard deviation (SD) and the median with range, whereas categorical variables are presented as frequencies and percentages. The study cohort was stratified by underlying disease etiology: Fuchs endothelial corneal dystrophy (FECD) and BK.

Univariable linear regression analyses were performed to identify potential predictors of postoperative visual acuity (measured in logarithm of the minimum angle of resolution [logMAR]). Explanatory variables included age, underlying disease, presence of cystoid macular edema (CME), epiretinal membrane (ERM), and wet age-related macular degeneration (AMD), macular comorbidity (defined as the presence of CME, ERM, or wet AMD), history of graft rejection, rebubbling, lens status, preoperative visual acuity, final endothelial cell density (ECD), and follow-up duration. For continuous explanatory variables, regression coefficients were standardized per one SD change to facilitate comparison across variables. To account for the correlation between eyes in bilateral cases, linear mixed-effects models were fitted using the lmer function in the lme4 package, with the patient included as a random effect. P values for fixed effects were calculated using Type III tests, and 95% confidence intervals (CIs) were estimated using the Wald method.

Subsequently, multivariable linear regression analyses were conducted to determine independent predictors of postoperative visual acuity. The full model incorporated all candidate explanatory variables that demonstrated potential associations in the univariable analyses. The final parsimonious model was selected based on the minimum Akaike Information Criterion (AIC) among all possible variable combinations. As in the univariable analysis, linear mixed-effects models (lme4 package) were utilized with the patient included as a random effect.

The relationship between preoperative and postoperative visual acuity was visualized using scatter plots stratified by macular comorbidity status. Linear regression lines were fitted to each subgroup, and a reference line indicating no change (y = x) was superimposed for visual comparison. The distribution of preoperative visual acuity between patients with and without macular comorbidities was compared using the Wilcoxon rank sum test and illustrated with box plots. This non-parametric approach was selected owing to the non-normal distribution of the visual acuity data. All statistical

analyses were performed using R version 4.4.3 (The R Foundation for Statistical Computing, Vienna, Austria). A two-sided P value of less than 0.05 was considered statistically significant.

## Results

The study included 77 eyes of 66 patients, comprising 55 unilateral and 11 bilateral cases (Table 1, S1 Table. Dataset). The mean age was 72.2 ± 9.2 years (median 74.0; range, 48–87 years), with a female predominance (68.8%). The underlying disease was FECD in 35 eyes (45.5%) and BK in 42 eyes (54.5%). The majority of eyes (92.2%) underwent primary DMEK, whereas 6.5% required repeat DMEK, and 1.3% underwent sequential DMEK/Descemet's stripping automated endothelial keratoplasty/DMEK procedures. All eyes were pseudophakic; 74 eyes (96.1%) had in-the-bag IOL implantation, and three eyes (3.9%) had scleral-fixated IOLs. Macular comorbidities were present in 32 eyes (41.6%), including CME in 12 (15.6%), ERM in 19 (24.7%), and wet AMD in four (5.2%). Graft rejection occurred in two eyes (2.6%), and rebubbling was required in 14 (18.2%). The mean preoperative visual acuity was 0.718 ± 0.508 logMAR (median 0.523; range, 0.05–2.00), which improved to 0.157 ± 0.246 logMAR (median 0.097; range, −0.08–1.52) postoperatively. The mean final ECD was 933.2 ± 390.2 cells/mm², and the mean follow-up period was 1461.0 ± 1011.4 days.

Univariable regression analysis revealed several factors significantly associated with postoperative visual acuity (Table 2). Older age was associated with worse postoperative visual acuity (coefficient, 0.087 logMAR per SD increase; 95% CI, 0.032–0.142; P = 0.002). The presence of CME was associated with poorer visual outcomes. Macular comorbidity was also significantly associated with worse postoperative visual acuity. Preoperative visual acuity demonstrated the strongest association, with a coefficient of 0.116 logMAR per SD increase (95% CI, 0.069–0.164; P < 0.001). Longer follow-up duration was associated with better postoperative visual acuity (coefficient, −0.074 logMAR per SD; 95% CI, −0.119 to −0.030; P = 0.002). No significant associations were observed for underlying disease type (P = 0.263), ERM (P = 0.325), wet AMD (P = 0.424), graft rejection (P = 0.600), rebubbling (P = 0.707), lens status (P = 0.237), or final ECD (P = 0.333).

The final AIC-optimized multivariable regression model identified two independent predictors of postoperative visual acuity (Table 3). Preoperative visual acuity remained strongly associated with postoperative visual acuity (coefficient, 0.114 logMAR per SD increase; 95% CI, 0.067–0.162; P < 0.001). The presence of macular comorbidity was independently associated

with worse postoperative visual acuity (coefficient, 0.112 logMAR; 95% CI, 0.014–0.211; P = 0.023). In the full multivariable model, neither age (P = 0.270) nor follow-up duration (P = 0.124) maintained statistical significance after adjustment for other covariates. Scatter plots of preoperative versus postoperative visual acuity stratified by macular comorbidity status demonstrated distinct patterns between the groups (Fig 1). In eyes without macular comorbidity (n = 45), the regression line showed a relatively flat slope, with most data points clustering near the lower range of postoperative visual acuity, indicating substantial visual improvement regardless of baseline vision. In contrast, eyes with macular comorbidity (n = 32) displayed a steeper regression slope, suggesting that postoperative visual outcomes in this subgroup were more dependent on preoperative visual acuity. Both subgroups showed data points predominantly below the reference line (y = x), indicating overall improvement in visual acuity following DMEK.

Preoperative visual acuity did not differ significantly between patients with and without macular comorbidities (Fig 2). The mean preoperative visual acuity was 0.814 ± 0.610 logMAR (median 0.523; range, 0.046–2.000) in patients with macular comorbidity and 0.650 ± 0.416 logMAR (median 0.523; range, 0.046–1.699) in those without (Wilcoxon rank sum test, P = 0.395). Notably, however, the macular comorbidity group included five eyes with a preoperative visual acuity of 2.0 logMAR, whereas the maximum value in the group without macular comorbidities was limited to 1.699 logMAR.

## Discussion

In this study, we investigated factors associated with postoperative visual acuity following endothelial keratoplasty (primarily DMEK) in our cohort. The present analysis focused on final visual acuity in eyes with a clear graft; corneal-specific

**Table 1. Patient demographics and baseline characteristics.**

| Item [unit] | Statistics Category | Overall (N = 77) | Disease | |
|---|---|---|---|---|
| | | | [1] FECD (N = 35) | [2] BK (N = 42) |
| Age [years] | Mean (SD) | 72.2 (9.2) | 71.3 (10.7) | 72.9 (7.9) |
| | Median [Range] | 74.0 [48, 87] | 73.0 [48, 87] | 74.0 [56, 86] |
| Sex | [1] Male | 24 (31.2%) | 11 (31.4%) | 13 (31.0%) |
| | [2] Female | 53 (68.8%) | 24 (68.6%) | 29 (69.0%) |
| Disease | [1] FECD | 35 (45.5%) | 35 (100.0%) | 0 (0.0%) |
| | [2] BK | 42 (54.5%) | 0 (0.0%) | 42 (100.0%) |
| Surgery type | [1] DMEK | 71 (92.2%) | 31 (88.6%) | 40 (95.2%) |
| | [2] DMEK/ DMEK | 5 (6.5%) | 3 (8.6%) | 2 (4.8%) |
| | [4] DMEK/ DSAEK/ DMEK | 1 (1.3%) | 1 (2.9%) | 0 (0.0%) |
| CME | [1] No | 65 (84.4%) | 32 (91.4%) | 33 (78.6%) |
| | [2] Yes | 12 (15.6%) | 3 (8.6%) | 9 (21.4%) |
| ERM | [1] No | 58 (75.3%) | 26 (74.3%) | 32 (76.2%) |
| | [2] Yes | 19 (24.7%) | 9 (25.7%) | 10 (23.8%) |
| Wet AMD | [1] No | 73 (94.8%) | 34 (97.1%) | 39 (92.9%) |
| | [2] Yes | 4 (5.2%) | 1 (2.9%) | 3 (7.1%) |
| Macular comorbidity | [1] No | 45 (58.4%) | 23 (65.7%) | 22 (52.4%) |
| | [2] Yes | 32 (41.6%) | 12 (34.3%) | 20 (47.6%) |
| Rejection | [1] No | 75 (97.4%) | 34 (97.1%) | 41 (97.6%) |
| | [2] Yes | 2 (2.6%) | 1 (2.9%) | 1 (2.4%) |
| Rebubbling | [1] No | 63 (81.8%) | 24 (68.6%) | 39 (92.9%) |
| | [2] Yes | 14 (18.2%) | 11 (31.4%) | 3 (7.1%) |
| Type of EK | [1] DMEK | 77 (100.0%) | 35 (100.0%) | 42 (100.0%) |
| Lens status | [1] IOL in the bag | 74 (96.1%) | 35 (100.0%) | 39 (92.9%) |
| | [2] Scleral fixation | 3 (3.9%) | 0 (0.0%) | 3 (7.1%) |
| Preoperative VA [logMAR] | Mean (SD) | 0.718 (0.508) | 0.632 (0.367) | 0.790 (0.596) |
| | Median [Range] | 0.523 [0.05, 2.00] | 0.523 [0.05, 1.70] | 0.523 [0.05, 2.00] |
| Final ECD | Mean (SD) | 933.2 (390.2) | 949.1 (444.8) | 919.6 (341.8) |
| | Median [Range] | 786.5 [360, 1934] | 746.0 [465, 1934] | 853.0 [360, 1730] |
| Follow up days from last surgery | Mean (SD) | 1461.0 (1011.4) | 1380.9 (1186.0) | 1527.8 (848.4) |
| | Median [Range] | 1314.0 [98, 4227] | 932.0 [212, 4227] | 1694.0 [98, 2938] |
| Postoperative VA [logMAR] | Mean (SD) | 0.157 (0.246) | 0.122 (0.180) | 0.185 (0.289) |
| | Median [Range] | 0.097 [-0.08, 1.52] | 0.046 [-0.08, 0.70] | 0.126 [-0.08, 1.52] |

The study population consists of 66 patients (55 unilateral and 11 bilateral cases). FECD, Fuchs endothelial corneal dystrophy; BK, bullous keratopathy; DMEK, Descemet membrane endothelial keratoplasty; DSAEK, Descemet stripping automated endothelial keratoplasty; CME, cystoid macular edema; ERM, epiretinal membrane; AMD, age-related macular degenerati on; EK, endothelial keratoplasty; VA, visual acuity; logMAR, logarithm of the minimum angle of resolution; ECD, endothelial cell density; IOL, intraocular lens; SD, standard deviation.

**Table 2. Univariate regression analysis for postoperative visual acuity [logMAR].**

| Explanatory variable | Category Unit | VA [logMAR] Mean (SD), N | Coefficient | | P value |
|---|---|---|---|---|---|
| | | | Estimate | | 95% C.I. |
| Age [years] | 1 SD | 0.157 (0.246), 77 | 0.087 | [0.032, 0.142] | 0.002 |
| Disease | [1] FECD | 0.122 (0.180), 35 | Reference | – | 0.263 |
| | [2] BK | 0.185 (0.289), 42 | 0.077 | [-0.050, 0.204] | |
| CME | [1] No | 0.121 (0.184), 65 | Reference | – | 0.003 |
| | [2] Yes | 0.348 (0.417), 12 | 0.123 | [-0.020, 0.266] | |
| ERM | [1] No | 0.141 (0.251), 58 | Reference | – | 0.325 |
| | [2] Yes | 0.205 (0.230), 19 | 0.068 | [-0.048, 0.184] | |
| Wet AMD | [1] No | 0.151 (0.238), 73 | Reference | – | 0.424 |
| | [2] Yes | 0.253 (0.400), 4 | 0.099 | [-0.167, 0.366] | |
| Macular comorbidity | [1] No | 0.092 (0.144), 45 | Reference | – | 0.005 |
| | [2] Yes | 0.248 (0.323), 32 | 0.092 | [-0.017, 0.201] | |
| Rejection | [1] No | 0.154 (0.248), 75 | Reference | – | 0.600 |
| | [2] Yes | 0.247 (0.213), 2 | 0.104 | [-0.090, 0.299] | |
| Rebubbling | [1] No | 0.162 (0.269), 63 | Reference | – | 0.707 |
| | [2] Yes | 0.134 (0.094), 14 | −0.047 | [-0.159, 0.066] | |
| Lens status | [1] IOL in the bag | 0.150 (0.237), 74 | Reference | – | 0.237 |
| | [2] Scleral fixation | 0.322 (0.460), 3 | 0.170 | [-0.133, 0.473] | |
| Preoperative VA [logMAR] | 1 SD | 0.157 (0.246), 77 | 0.116 | [0.069, 0.164] | < 0.001 |
| Final ECD [cells/mm2] | 1 SD | 0.155 (0.247), 76 | −0.018 | [-0.057, 0.020] | 0.333 |
| Follow-up days | 1 SD | 0.157 (0.246), 77 | −0.074 | [-0.119, -0.030] | 0.002 |

Standard deviations (SDs) for continuous exploratory variables: Age = 9.2 years, Preoperative VA = 0.508 logMAR, Final ECD = 390.2 cells/mm2, Follow-up days = 1011.4 days.

The analysis population consists of 66 patients (55 unilateral and 11 bilateral cases). To account for the correlation between eyes in bilateral cases, a linear mixed-effects model was used with the patient as a random effect.

VA, visual acuity; logMAR, logarithm of the minimum angle of resolution; FECD, Fuchs endothelial corneal dystrophy; BK, bullous keratopathy; CME, cystoid macular edema; ERM, epiretinal membrane; AMD, age-related macular degeneration; ECD, endothelial cell density; IOL, intraocular lens; CI, confidence interval; SD, standard deviation.

**Table 3. Multivariate regression analysis for postoperative visual acuity [logMAR].**

| Explanatory variable | Category Unit | Full model | | | Final model | | |
|---|---|---|---|---|---|---|---|
| | | Coefficient | 95% C.I. | P value | Coefficient | 95% C.I. | P value |
| Age [years] | 1 SD | 0.033 | [-0.027, 0.094] | 0.270 | | | |
| Macular comorbidity | [1] No | Reference | – | 0.044 | Reference | – | 0.023 |
| | [2] Yes | 0.102 | [0.001, 0.203] | | 0.112 | [0.014, 0.211] | |
| Preoperative VA [logMAR] | 1 SD | 0.078 | [0.030, 0.125] | < 0.001 | 0.114 | [0.067, 0.162] | < 0.001 |
| Follow-up days | 1 SD | −0.042 | [-0.096, 0.012] | 0.124 | | | |

Full model includes all explanatory variables. Final model includes only variables achieving the minimum AIC in all combinations of explanatory variables.

Standard deviations (SDs) for continuous exploratory variables: Age = 9.2 years, Preoperative VA = 0.508 logMAR, Final ECD = 390.2 cells/mm2, Follow-up days = 1011.4 days.

The analysis population consists of 66 patients (55 unilateral and 11 bilateral cases). To account for the correlation between eyes in bilateral cases, a linear mixed-effects model was used with the patient as a random effect. k sum test, P = 0.395.

VA, visual acuity; logMAR, logarithm of the minimum angle of resolution; CI, confidence interval; AIC, Akaike information criterion; SD, standard deviation.

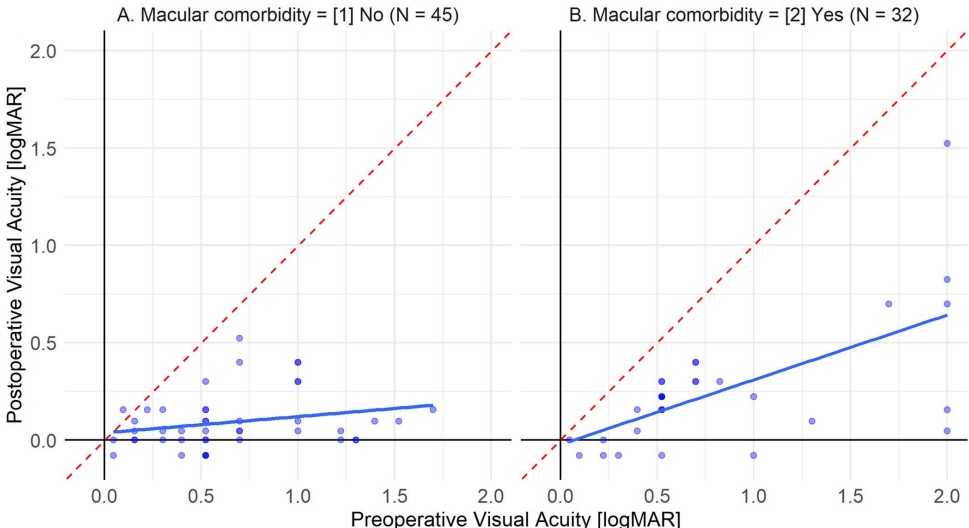

**Fig 1. Scatter plots of pre- vs. post-operative visual acuity [logMAR] by macular comorbidity status.** Scatter plots of preoperative versus postoperative visual acuity stratified by macular comorbidity status demonstrated distinct patterns between the groups (Fig 1). In eyes without macular comorbidity (N = 45), the regression line showed a relatively flat slope, with most data points clustering near the lower range of postoperative visual acuity, indicating substantial visual improvement regardless of baseline vision. In contrast, eyes with macular comorbidity (N = 32) displayed a steeper regression slope, suggesting that postoperative visual outcomes in this subgroup were more dependent on preoperative visual acuity. Both subgroups showed data points predominantly below the reference line (y = x), indicating overall improvement in visual acuity following DMEK.

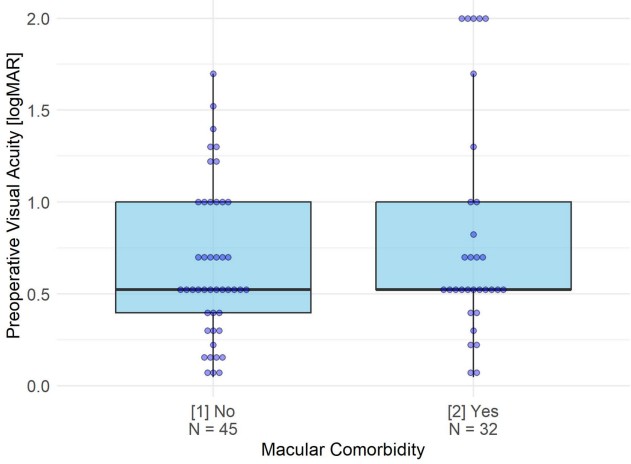

**Fig 2. Box-whisker plot of preoperative visual acuity [logMAR] by macular comorbidity status.** Comparison of preoperative visual acuity between patients with and without macular comorbidity revealed no statistically significant difference (Fig 2). The mean preoperative visual acuity was 0.650 ± 0.416 logMAR (median 0.523; range, 0.046–1.699) in patients without macular comorbidity and 0.814 ± 0.610 logMAR (median 0.523; range, 0.046–2.000) in those with macular comorbidity (Wilcoxon rank sum test, P = 0.395). Notably, however, the macular comorbidity group included 5 eyes with a preoperative visual acuity of 2.0 logMAR, whereas the maximum value in the group without macular comorbidity was limited to 1.699 logMAR.

postoperative parameters such as ECD or transient graft detachment were not included, as they primarily reflect graft status rather than visual acuity once corneal transparency is maintained.

Our analysis revealed that preoperative visual acuity and the presence of macular comorbidities were significantly associated with worse postoperative visual acuity, whereas other ocular comorbidities such as CME, ERM, or wet AMD did not show significant effects in the multivariable model. Previous large-scale studies have consistently shown that poorer preoperative visual acuity is among the strongest predictors of suboptimal postoperative outcomes after endothelial keratoplasty [14]. These reports suggest that poorer preoperative visual acuity reflects the combined influence of corneal pathology and concomitant retinal disease on final visual outcomes.

Although fewer studies have focused on macular disease in the context of corneal transplantation, it is intuitive that comorbid macular pathology limits the functional visual gain postoperatively even when the corneal clarity is restored.

Although CME, ERM, and wet AMD showed trends toward poorer visual outcomes in univariable analyses, none remained statistically significant in multivariable models. This suggests that, while these factors may contribute to vision impairment, they are overshadowed by the stronger impact of preoperative visual acuity and macular comorbidity in our dataset. Careful preoperative screening for subtle macular changes remains essential.

Consistent with previous reports, our findings support the concept that concomitant macular disease limits functional visual recovery after corneal transplantation, even when corneal clarity is restored [15–17]. However, as mentioned earlier, determining the presence of macular disease is difficult in some cases owing to corneal opacity caused by corneal disease. Thus, for cases with poor preoperative visual acuity in which macular examination is not possible, treatment should be planned considering the possibility of macular disease.

This study has some limitations. It was conducted retrospectively at a few centers with a relatively small sample size. Additionally, our analysis focused primarily on postoperative visual acuity and did not fully address other parameters such as ECD or long-term graft survival. Preoperative macular status could not be reliably determined in all cases because of severe corneal opacity, and therefore, some macular comorbidities detected postoperatively might have been pre-existing but unrecognized. Further prospective, multicenter studies with extended follow-up are warranted to clarify the complex pathophysiologic interplay between corneal graft integrity and retinal comorbidities. Because macular diseases may influence visual acuity differently depending on their stage and the condition of the retinal pigment epithelium, future investigations should include detailed retinal evaluations to better understand their impact on postoperative visual outcomes.

In conclusion, we identified preoperative visual acuity and macular comorbidity as independent risk factors for poor postoperative visual acuity following endothelial keratoplasty. These factors should be considered when planning and counselling for corneal transplantation, and a comprehensive preoperative evaluation is crucial.

## Supporting information

**S1 Table. Dataset. The dataset includes all information used for this study.**
(PDF)

## Acknowledgments

We thank Eisei Oda, PhD, of StatLink Consulting, for providing statistical advice on revising the tables, figures, and manuscript.

## Author contributions

**Conceptualization:** Takahiko Hayashi.

**Data curation:** Naoya Nakagawa, Ami Igarashi, Hideaki Yokogawa, Takahiko Hayashi.

**Formal analysis:** Hideaki Yokogawa, Takahiko Hayashi.

**Investigation:** Ami Igarashi, Akira Kobayashi, Tomomi Higashide, Satoru Yamagami, Takahiko Hayashi.

**Methodology:** Naoya Nakagawa, Ami Igarashi, Akira Kobayashi, Takahiko Hayashi.

**Project administration:** Hideaki Yokogawa, Akira Kobayashi, Satoru Yamagami, Takahiko Hayashi.

**Resources:** Hideaki Yokogawa, Takahiko Hayashi.

**Software:** Takahiko Hayashi.

**Supervision:** Tomomi Higashide, Satoru Yamagami, Takahiko Hayashi.

**Validation:** Takahiko Hayashi.

**Visualization:** Takahiko Hayashi.

**Writing – original draft:** Naoya Nakagawa, Ami Igarashi, Takahiko Hayashi.

**Writing – review & editing:** Akira Kobayashi, Tomomi Higashide, Satoru Yamagami, Takahiko Hayashi.

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
