## [Decision Letter · Decision Letter 0]

7 Jan 2026

PONE-D-25-65215Impact of postoperative macular disorders on visual outcomes after Descemet’s membrane endothelial keratoplasty: A multicenter analysisPLOS One

Dear Dr. Hayashi,

Thank you for submitting your manuscript to PLOS ONE. After careful consideration, we feel that it has merit but does not fully meet PLOS ONE’s publication criteria as it currently stands. Therefore, we invite you to submit a revised version of the manuscript that addresses the points raised during the review process.

We look forward to receiving your revised manuscript.

Kind regards,

Hidenaga Kobashi, M.D., Ph.D.

Academic Editor

PLOS One

Journal Requirements:

2. We note that there is identifying data in the Supporting Information file <data set.pdf >. Due to the inclusion of these potentially identifying data, we have removed this file from your file inventory. Prior to sharing human research participant data, authors should consult with an ethics committee to ensure data are shared in accordance with participant consent and all applicable local laws.

-Location data

Please remove or anonymize all personal information (<age>), ensure that the data shared are in accordance with participant consent, and re-upload a fully anonymized data set. Please note that spreadsheet columns with personal information must be removed and not hidden as all hidden columns will appear in the published file.

Additional Editor Comments:

The reviewers and editor have completed their assessments of your manuscript

Impact of postoperative macular disorders on visual outcomes after Descemet’s membrane endothelial keratoplasty: A multicenter analysis (PONE-D-25-65215)

and would like to publish it in the journal once you have responded to the referees' comments (enclosed below). In the cover letter with the revised manuscript, please indicate how each of the reviewers' suggestions was addressed.

Reviewers' comments:

Reviewer's Responses to Questions

**Comments to the Author**

1. Is the manuscript technically sound, and do the data support the conclusions?

Reviewer #1: No

Reviewer #2: Yes

2. Has the statistical analysis been performed appropriately and rigorously?

Reviewer #1: No

Reviewer #2: Yes

3. Have the authors made all data underlying the findings in their manuscript fully available?

Reviewer #1: Yes

Reviewer #2: Yes

4. Is the manuscript presented in an intelligible fashion and written in standard English?

Reviewer #1: Yes

Reviewer #2: Yes

5. Review Comments to the Author

Reviewer #1: The authors present an interesting study that analyzes whether postOP macular disorders limit final visual recovery after DMEK. Overall, the manuscript is readable and the clinical message is clear and sensible: macular disease matters. However, several aspects of the study currently make the manuscript insufficient for publication due to weak strength of evidence supporting their conclusions. The main limitations are: 1) the authors do not state whether any of the listed macular disorders were pre-existing prior to surgery. This is a very important point that needs to be addressed, in addition to stating whether there was any macular edema or a dry/wet AMD preoperatively. As the authors acknowledge in limitations, the corneal opacity limiting a proper preOP OCT scan analysis highlights this loophole of information. 2) the authors should address the non-independence from including both eyes of some patients without accounting for clustering, or restrict to one eye per patient and redo analyses as sensitivity. At minimum, the authors should report how many cases were bilateral and include a sensitivity analysis. 3) the authors model CME, ERM, wet AMD, and "macular comorbidity" (defined as "any of CME/ERM/wet AMD"), which is mathematically redundant and invites collinearity/suppression effects. The authors should either model each individual macular diagnosis and not include the composite, or model only the composite and do not include the individual diagnoses. 4) The primary endpoint is BCVA at final follow-up; however, the follow-up time ranges from 12 to 4302 days (!). Given that the final VA strongly depends on the time since surgery and whether macular events occurred early/late, the authors should define a more standardized time point (e.g., 6 months or 12 months) or include follow-up duration (or time since surgery at which the final VA was measured) as an adjustment variable. 5) Several fundamental variables known to affect VA after DMEK are omitted, including baseline BCVA, lens status (phakic/pseudophakic), DMEK vs. triple-DMEK, rebubbling rates, ECC of the graft, and corneal comorbidity severity (krachmer grade). 6) the authors state that 47% of patients were FECD and 41% BK; what about the rest?

Overall, while the study question is worthwhile, the authors should address the aforementioned points to be considered for publication in PLOS one

Reviewer #2: Dear Authors,

Thanks for this manuscript, which addresses a clinically relevant and timely question: the impact of postoperative macular disorders on visual outcomes following DMEK. The manuscript is generally well written, the surgical technique appears standardized, and the authors appropriately attempt multivariate modeling to identify independent predictors of postoperative visual acuity.

Clarification is needed regarding the timing and definition of macular comorbidities, the robustness of the statistical modeling given small subgroup sizes, and the clinical interpretation of statistically significant findings. Addressing these points would substantially improve the manuscript.

Major comments

1. Definition and timing of macular comorbidity

Macular comorbidity is defined as the presence of CME, ERM, or wet AMD detected during postoperative follow-up. It is unclear whether these conditions were pre-existing but only detected postoperatively due to improved media clarity, or whether they truly developed after surgery. This distinction is critical for causal interpretation and clinical counselling. The authors should clarify whether preoperative OCT was routinely performed and, if possible, distinguish between pre-existing and de novo postoperative macular disease. If this is not feasible, the limitation should be explicitly acknowledged in the Discussion.

2. Statistical modeling and risk of overfitting

The multivariate linear regression includes multiple covariates despite a relatively small sample size (82 eyes) and very small subgroups (e.g., wet AMD n=4, PEX n=2). This raises concerns regarding model stability and overfitting, particularly in the context of AIC-based variable selection. The authors are encouraged to justify the model complexity (e.g., events-per-variable considerations) or perform sensitivity analyses excluding very small subgroups.

3. Interpretation and clinical relevance of regression coefficients

Although several predictors reach statistical significance, the clinical magnitude of some effects appears modest. For example, the age coefficient (β ≈ 0.006 logMAR per year) represents a small change in vision over a decade. The authors should consider translating key coefficients into clinically interpretable terms (e.g., Snellen equivalents or lines of vision) also in the discussion and discuss whether these differences exceed commonly accepted thresholds for clinical relevance.

4. Handling and interpretation of pseudoexfoliation syndrome

PEX is identified as a strong independent predictor of poor postoperative visual acuity; however, only two PEX cases are included. While the sensitivity analysis excluding PEX is appropriate, conclusions regarding PEX as a high-risk subgroup should be tempered. The authors should more clearly acknowledge the uncertainty associated with this very small sample size and frame the findings as hypothesis-generating rather than definitive.

5. Absence of corneal-specific postoperative parameters

The analysis focuses exclusively on postoperative visual acuity without incorporating corneal-specific outcomes such as endothelial cell density, graft detachment or rebubbling, graft clarity, or rejection episodes. These factors can independently influence visual outcomes and may correlate with age or PEX. The rationale for excluding these variables should be clarified, and this limitation should be more explicitly discussed.

Minor comments

1. Follow-up duration is highly variable. Please clarify whether visual acuity was assessed at a standardized postoperative time point or whether time from surgery to assessment was considered in the analysis.

2. Terminology alternates between macular disease, macular disorder, and macular comorbidity. Consistent terminology throughout the manuscript would improve clarity.

3. For CME cases, it would be helpful to report whether CME resolved with treatment and whether final visual acuity was measured after resolution, as this directly affects interpretation of the results.

4. Figures 1 and 2 provide limited additional information beyond the regression analyses. Consider simplifying the figures or enhancing them with stratified regression lines and confidence intervals.

6. PLOS authors have the option to publish the peer review history of their article (what does this mean?). If published, this will include your full peer review and any attached files.

Reviewer #1: No

Reviewer #2: No

---

## [Author Response · Author response to Decision Letter 1]

8 Feb 2026

Additional Editor Comments:

The reviewers and editor have completed their assessments of your manuscript

Impact of postoperative macular disorders on visual outcomes after Descemet’s membrane endothelial keratoplasty: A multicenter analysis (PONE-D-25-65215)

and would like to publish it in the journal once you have responded to the referees' comments (enclosed below). In the cover letter with the revised manuscript, please indicate how each of the reviewers' suggestions was addressed.

Reviewers' comments:

Reviewer's Responses to Questions

5. Review Comments to the Author

Reviewer #1: The authors present an interesting study that analyzes whether postOP macular disorders limit final visual recovery after DMEK. Overall, the manuscript is readable and the clinical message is clear and sensible: macular disease matters. However, several aspects of the study currently make the manuscript insufficient for publication due to weak strength of evidence supporting their conclusions.

The main limitations are: 1) the authors do not state whether any of the listed macular disorders were pre-existing prior to surgery. This is a very important point that needs to be addressed, in addition to stating whether there was any macular edema or a dry/wet AMD preoperatively. As the authors acknowledge in limitations, the corneal opacity limiting a proper preOP OCT scan analysis highlights this loophole of information.

Response:

We thank the reviewer for this important observation. In many cases, severe corneal edema precluded reliable preoperative fundus examination and optical coherence tomography (OCT), making it difficult to determine whether macular disorders were present preoperatively or developed de novo after surgery. This limitation has been explicitly acknowledged in the Discussion section.

Revision:

Discussion, Lines 273–275

“Preoperative macular status could not be reliably determined in all cases because of severe corneal opacity, and therefore, some macular comorbidities detected postoperatively might have been pre-existing but unrecognized.”

2) the authors should address the non-independence from including both eyes of some patients without accounting for clustering, or restrict to one eye per patient and redo analyses as sensitivity. At minimum, the authors should report how many cases were bilateral and include a sensitivity analysis.

Response:

We agree with this concern. We reanalyzed the data using linear mixed-effects models with the patient included as a random effect to account for intrapatient correlation. The number of bilateral cases has been clearly reported, and all tables and figures have been updated accordingly.

Revision:

Statistical Methods, Lines 119–121

“To account for the correlation between eyes in bilateral cases, linear mixed-effects models were fitted using the lmer function in the lme4 package, with the patient included as a random effect.”

Results, Lines 141–142

“The study included 77 eyes of 66 patients, comprising 55 unilateral and 11 bilateral cases (Table 1).”

Tables 1–3 and Figures 1 and 2 revised accordingly.

3) the authors model CME, ERM, wet AMD, and "macular comorbidity" (defined as "any of CME/ERM/wet AMD"), which is mathematically redundant and invites collinearity/suppression effects. The authors should either model each individual macular diagnosis and not include the composite, or model only the composite and do not include the individual diagnoses.

Response:

We appreciate this methodological suggestion. To avoid collinearity and overfitting, CME, ERM, and wet AMD were excluded from the multivariable model, and only the composite variable “macular comorbidity” was retained. Sensitivity analyses excluding individual diagnoses were also performed.

Revision:

Statistical Methods, Lines 124–128

“Subsequently, multivariable linear regression analyses were conducted to determine independent predictors of postoperative visual acuity. The full model incorporated all candidate explanatory variables that demonstrated potential associations in the univariable analyses. The final parsimonious model was selected based on the minimum Akaike Information Criterion (AIC) among all possible variable combinations.”

Tables 2 and 3 revised accordingly.

4) The primary endpoint is BCVA at final follow-up; however, the follow-up time ranges from 12 to 4302 days (!). Given that the final VA strongly depends on the time since surgery and whether macular events occurred early/late, the authors should define a more standardized time point (e.g., 6 months or 12 months) or include follow-up duration (or time since surgery at which the final VA was measured) as an adjustment variable.

Response:

We agree with the reviewer. We included follow-up duration as a covariate in the regression analyses. Based on these analyses, we have revised the tables and figures.

Revision:

Statistical Methods, Lines 114–118

“Explanatory variables included age, underlying disease, presence of cystoid macular edema (CME), epiretinal membrane (ERM), and wet age-related macular degeneration (AMD), macular comorbidity (defined as the presence of CME, ERM, or wet AMD), history of graft rejection, rebubbling, lens status, preoperative visual acuity, final endothelial cell density (ECD), and follow-up duration.”

Results, Lines 191–192

“In the full multivariable model, neither age (P = 0.270) nor follow-up duration (P = 0.124) maintained statistical significance after adjustment for other covariates.”

Tables 2 and 3 revised.

Figures 1 and 2 updated.

5) Several fundamental variables known to affect VA after DMEK are omitted, including baseline BCVA, lens status (phakic/pseudophakic), DMEK vs. triple-DMEK, rebubbling rates, ECC of the graft, and corneal comorbidity severity (krachmer grade).

Response:

The purpose of this study was to specifically examine the relationship between postoperative visual acuity and macular disease. Therefore, only variables considered statistically essential for this objective were included. Nevertheless, lens status, rebubbling, and preoperative visual acuity have been added to the tables to improve transparency.

Revision:

Statistical Methods, Lines 115–119

“Explanatory variables included age, underlying disease, presence of cystoid macular edema (CME), epiretinal membrane (ERM), and wet age-related macular degeneration (AMD), macular comorbidity (defined as the presence of CME, ERM, or wet AMD), history of graft rejection, rebubbling, lens status, preoperative visual acuity, final endothelial cell density (ECD), and follow-up duration.”

6) the authors state that 47% of patients were FECD and 41% BK; what about the rest? Overall, while the study question is worthwhile, the authors should address the aforementioned points to be considered for publication in PLOS one

Response:

All remaining cases were reclassified as bullous keratopathy (BK); the text has been corrected accordingly.

Revision:

Results, Lines 143–146

“The underlying disease was FECD in 35 eyes (45.5%) and BK in 42 eyes (54.5%). The majority of eyes (92.2%) underwent primary DMEK, whereas 6.5% required repeat DMEK, and 1.3% underwent sequential DMEK/Descemet's stripping automated endothelial keratoplasty/DMEK procedures.”

Table 1 revised.

Reviewer #2: Dear Authors,

Thanks for this manuscript, which addresses a clinically relevant and timely question: the impact of postoperative macular disorders on visual outcomes following DMEK. The manuscript is generally well written, the surgical technique appears standardized, and the authors appropriately attempt multivariate modeling to identify independent predictors of postoperative visual acuity.

Clarification is needed regarding the timing and definition of macular comorbidities, the robustness of the statistical modeling given small subgroup sizes, and the clinical interpretation of statistically significant findings. Addressing these points would substantially improve the manuscript.

Major comments

1. Definition and timing of macular comorbidity

Macular comorbidity is defined as the presence of CME, ERM, or wet AMD detected during postoperative follow-up. It is unclear whether these conditions were pre-existing but only detected postoperatively due to improved media clarity, or whether they truly developed after surgery. This distinction is critical for causal interpretation and clinical counselling. The authors should clarify whether preoperative OCT was routinely performed and, if possible, distinguish between pre-existing and de novo postoperative macular disease. If this is not feasible, the limitation should be explicitly acknowledged in the Discussion.

Response:

As noted above, preoperative differentiation was not always feasible because of corneal opacity. This limitation has been clearly acknowledged in the Discussion section.

Revision:

Discussion, Lines 273–275

“Preoperative macular status could not be reliably determined in all cases because of severe corneal opacity, and therefore, some macular comorbidities detected postoperatively might have been pre-existing but unrecognized.”

2. Statistical modeling and risk of overfitting

The multivariate linear regression includes multiple covariates despite a relatively small sample size (82 eyes) and very small subgroups (e.g., wet AMD n=4, PEX n=2). This raises concerns regarding model stability and overfitting, particularly in the context of AIC-based variable selection. The authors are encouraged to justify the model complexity (e.g., events-per-variable considerations) or perform sensitivity analyses excluding very small subgroups.

Response:

Small subgroups such as PEX and wet AMD were excluded from the multivariable model to reduce overfitting and improve model stability.

Revision:

Statistical Methods, Lines 114–118

“Explanatory variables included age, underlying disease, presence of cystoid macular edema (CME), epiretinal membrane (ERM), and wet age-related macular degeneration (AMD), macular comorbidity (defined as the presence of CME, ERM, or wet AMD), history of graft rejection, rebubbling, lens status, preoperative visual acuity, final endothelial cell density (ECD), and follow-up duration.”

Results, Lines 143–144

“The underlying disease was FECD in 35 eyes (45.5%) and BK in 42 eyes (54.5%).”

3. Interpretation and clinical relevance of regression coefficients

Although several predictors reach statistical significance, the clinical magnitude of some effects appears modest. For example, the age coefficient (β ≈ 0.006 logMAR per year) represents a small change in vision over a decade. The authors should consider translating key coefficients into clinically interpretable terms (e.g., Snellen equivalents or lines of vision) also in the discussion and discuss whether these differences exceed commonly accepted thresholds for clinical relevance.

Response:

Following additional analyses, age was no longer retained in the final multivariable model. The Discussion section has been revised to focus on clinically meaningful predictors—preoperative visual acuity and macular comorbidity—without overemphasizing statistically significant but clinically modest effects.

Revision:

Results, Lines 187–201

“Preoperative visual acuity remained strongly associated with postoperative visual acuity (coefficient, 0.114 logMAR per SD increase; 95% CI, 0.067–0.162; P < 0.001). The presence of macular comorbidity was independently associated

with worse postoperative visual acuity (coefficient, 0.112 logMAR; 95% CI, 0.014–0.211; P = 0.023). In the full multivariable model, neither age (P = 0.270) nor follow-up duration (P = 0.124) maintained statistical significance after adjustment for other covariates. Scatter plots of preoperative versus postoperative visual acuity stratified by macular comorbidity status demonstrated distinct patterns between the groups (Fig 1). In eyes without macular comorbidity (n = 45), the regression line showed a relatively flat slope, with most data points clustering near the lower range of postoperative visual acuity, indicating substantial visual improvement regardless of baseline vision. In contrast, eyes with macular comorbidity (n = 32) displayed a steeper regression slope, suggesting that postoperative visual outcomes in this subgroup were more dependent on preoperative visual acuity. Both subgroups showed data points predominantly below the reference line (y = x), indicating overall improvement in visual acuity following DMEK.”

Discussion, Lines 248–253

“Our analysis revealed that preoperative visual acuity and the presence of macular comorbidities were significantly associated with worse postoperative visual acuity, whereas other ocular comorbidities such as CME, ERM, or wet AMD did not show significant effects in the multivariable model. Previous large-scale studies have consistently shown that poorer preoperative visual acuity is among the strongest predictors of suboptimal postoperative outcomes after endothelial keratoplasty [14].”

4. Handling and interpretation of pseudoexfoliation syndrome

PEX is identified as a strong independent predictor of poor postoperative visual acuity; however, only two PEX cases are included. While the sensitivity analysis excluding PEX is appropriate, conclusions regarding PEX as a high-risk subgroup should be tempered. The authors should more clearly acknowledge the uncertainty associated with this very small sample size and frame the findings as hypothesis-generating rather than definitive.

Response:

PEX cases were excluded from the final multivariable analysis, and related interpretations have been omitted to avoid overstatement.

Revision:

Statistical Methods, Lines 45–46 in the original manuscript

“Subgroup analyses were conducted after excluding pseudoexfoliation syndrome (PEX) cases to assess robustness.”

5. Absence of corneal-specific postoperative parameters

The analysis focuses exclusively on postoperative visual acuity without incorporating corneal-specific outcomes such as endothelial cell density, graft detachment or rebubbling, graft clarity, or rejection episodes. These factors can independently influence visual outcomes and may correlate with age or PEX. The rationale for excluding these variables should be clarified, and this limitation should be more explicitly discussed.

Response:

We fully agree with the reviewer. Following additional analyses, we included the suggested corneal-specific parameters in the regression analysis.

Revision:

Methods, Lines 114–123

“Explanatory variables included age, underlying disease, presence of cystoid macular edema (CME), epiretinal membrane (ERM), and wet age-related macular degeneration (AMD), macular comorbidity (defined as the presence of CME, ERM, or wet AMD), history of graft rejection, rebubbling, lens status, preoperative visual acuity, final endothelial cell density (ECD), and follow-up duration. For continuous explanatory variables, regression coefficients were standardized per one SD change to facilitate comparison across variables. To account for the correlation between eyes in bilateral cases, linear mixed-effects models were fitted using the lmer function in the lme4 package, with the patient included as a random effect. P values for fixed effects were calculated using Type III tests, and 95% confidence intervals (CIs) were estimated using the Wald method.”

Minor comments

1. Follow-up duration is highly variable. Please clarify whether visual acuity was assessed at a standardized postoperative time point or whether time from surgery to assessment was considered in the analysis.

Response:

We fully agree with the reviewer. Following your comment, we conducted an additional analysis to validate the variable “follow-up duration” as follows:

Addressed by inclusion as a covariate:

Results, Lines 169–173

“Longer follow-up duration was associated with better postoperative visual acuity (coefficient, −0.074 logMAR per S

---

## [Decision Letter · Decision Letter 1]

13 Feb 2026

Impact of postoperative macular comorbidity on visual outcomes after Descemet’s membrane endothelial keratoplasty: A multicenter analysis

PONE-D-25-65215R1

Dear Dr. Hayashi,

We’re pleased to inform you that your manuscript has been judged scientifically suitable for publication and will be formally accepted for publication once it meets all outstanding technical requirements.

Kind regards,

Hidenaga Kobashi, M.D., Ph.D.

Academic Editor

PLOS One

Additional Editor Comments (optional):

Reviewers' comments:

Reviewer's Responses to Questions

**Comments to the Author**

1. If the authors have adequately addressed your comments raised in a previous round of review and you feel that this manuscript is now acceptable for publication, you may indicate that here to bypass the “Comments to the Author” section, enter your conflict of interest statement in the “Confidential to Editor” section, and submit your "Accept" recommendation.

Reviewer #2: All comments have been addressed

2. Is the manuscript technically sound, and do the data support the conclusions?

Reviewer #2: Yes

3. Has the statistical analysis been performed appropriately and rigorously?

Reviewer #2: Yes

4. Have the authors made all data underlying the findings in their manuscript fully available?

Reviewer #2: No

5. Is the manuscript presented in an intelligible fashion and written in standard English?

Reviewer #2: Yes

6. Review Comments to the Author

Reviewer #2: (No Response)

7. PLOS authors have the option to publish the peer review history of their article (what does this mean?). If published, this will include your full peer review and any attached files.

Reviewer #2: No

---

## [Editor Report · Acceptance letter]

PONE-D-25-65215R1

PLOS One

Dear Dr. Hayashi,

I'm pleased to inform you that your manuscript has been deemed suitable for publication in PLOS One. Congratulations! Your manuscript is now being handed over to our production team.

Kind regards,

on behalf of

Dr. Hidenaga Kobashi

Academic Editor

PLOS One